# OODEEL: A HOLISTIC LIBRARY FOR UNIFIED POST-HOC OOD DETECTION RESEARCH AND APPLICATION

## ABSTRACT

We present OODEEL, an open-source Post-hoc Out-of-Distribution (OOD) detection library. OODEEL is designed as a highly customizable tool that supports a wide range of OOD detectors and can be applied to any model classifier architectures from both PyTorch and TensorFlow. It implements unified abstractions so that every building block, such as activation shaping and layer-wise aggregation, can be used seamlessly by any detector. It also provides a user-friendly API that allows for easy integration of new OOD detectors, which can then benefit from all these building blocks, and native compatibility with most TensorFlow and PyTorch models. OODEEL seamlessly handles standard OOD evaluation settings for benchmarking, including multiple ID/OOD datasets (both near- and far-OOD). Hence, we leverage its holistic implementation to address several critical aspects of OOD evaluation that are often overlooked in current benchmarks: robustness to model variability, effect of aggregation of layer-wise scores, effect of activation shaping, and link between in-distribution accuracy and OOD detection performances.

## 1 INTRODUCTION

Out-of-distribution (OOD) detection has become increasingly crucial for deploying robust machine learning systems in real-world applications. Neural network models, although effective, often exhibit overconfidence when encountering inputs outside their training distribution, which can lead to significant safety and reliability issues. To mitigate this, post-hoc OOD detection methods have emerged as attractive solutions, as they require no retraining and can readily be applied to existing pretrained models. These approaches typically operate by assigning OOD scores to inputs based on either model logits or internal layer feature values.

Some libraries, such as OpenOOD Zhang et al. (2023) and PyTorch-OOD Kirchheim et al. (2022), have been developed to foster research, benchmarking, and adoption of OOD detection. Despite their popularity, these libraries have notable limitations. Firstly, they implement OOD detectors separately, often ignoring the potential benefits of combining components such as activation shaping Sun et al. (2021); Xu et al. (2023); Djurisic et al. (2022) and layer-wise score aggregation Himmi et al. (2024); Dadalto et al. (2024); Haroush et al. (2022) across different OOD methods. Secondly, these libraries tend to restrict their usability to predefined models, neglecting the broader applicability of post-hoc methods to arbitrary user-provided classifiers. Lastly, they are generally restricted to a single deep learning framework, typically PyTorch, excluding TensorFlow practitioners from easy access to state-of-the-art OOD detection tools.

In this paper, we introduce OODEEL, an open-source, highly flexible, and holistic library that addresses these limitations. By supporting seamless integration with arbitrary pretrained models from both PyTorch and TensorFlow, it provides a unified implementation of post-hoc OOD detectors and components such as activation shaping and layer-wise aggregation. Moreover, its intuitive and unified API enables effortless application to many OOD detection test cases. OODEEL also includes a benchmarking functionality that allows for easy testing of implemented OOD detectors on the OpenOOD benchmark.

Leveraging the versatility of OODEEL, we conduct extensive experiments to rigorously assess several underexplored aspects of OOD detection, including the sensitivity of methods to model variability, the impact of aggregating scores across multiple layers, and the influence of activation shaping strategies. We also provide additional insights from our experiments about the link between ID accuracy and

AUROC, disparate performances on small-scale and large-scale datasets, and correlation between near and far OOD. Our findings not only reveal critical insights into the performance and robustness of current OOD methods but also highlight the practical advantages of a unified approach in developing and evaluating future OOD detection techniques. OODEEL[1] is available on GitHub, and we release an accompanying platform[2] for a comprehensive visualization of experiment results.

## 2 BACKGROUND

Out-of-distribution detection can be interpreted as a binary classification problem. The most common way of constructing an OOD detector is by using an OOD scoring function $s : \mathcal{X} \to \mathbb{R}$ and a threshold $\tau \in \mathbb{R}$:

$$\begin{cases} x \text{ predicted ID} & \text{if } s(x) \leq \tau, \\ x \text{ predicted OOD} & \text{if } s(x) > \tau. \end{cases}$$

Throughout the paper we adopt the convention that OOD scoring functions are designed so that they give higher scores to OOD data and lower scores to ID data. Given a pair of datasets $\mathcal{D}^-$ for ID data and $\mathcal{D}^+$ for OOD data, and an OOD detector $(s, \tau)$, we can define the following metrics:

- *False Positive Rate (FPR)*. The fraction of ID data that is incorrectly classified as being OOD, i.e. $FPR(\tau) = |\{x \in \mathcal{D}^- : s(x) > \tau\}|/|\mathcal{D}^-|$.

- *True Positive Rate (TPR)*. The fraction of OOD data that is correctly classified as being OOD, i.e. $TPR(\tau) = |\{x \in \mathcal{D}^+ : s(x) > \tau\}|/|\mathcal{D}^+|$.

In the context of OOD benchmarking, it is customary to measure the separation power of a scoring function $s$ independently of the chosen threshold $\tau$. The most popular such metric is the *AUROC* or area under the ROC curve, which is the parametric curve given by $\{(FPR(\tau), TPR(\tau)), \tau \in \mathbb{R}\}$.

Classical benchmarks in OOD research evaluate a score $s$ by computing its AUROC with respect to a bunch of different pairs of ID/OOD datasets, where the ID/OOD pairs are usually chosen so that they have semantically non-overlapping classes.

### 2.1 POST-HOC OOD

Post-hoc OOD is a popular class of OOD detection methods (the library OpenOOD Zhang et al. (2023) implements 24) that apply to already trained models without requiring training or fine-tuning. Let $f$ denote a classifier neural network whose weights have already been fixed through a training process. We assume the classifier $f$ is obtained as a composition of different parametric functions called *layers*,

$$f(x) = h^{\ell} \circ h^{\ell-1} \circ \cdots \circ h^1(x).$$

Assume that the values $f(x) \in \mathbb{R}^{d_\ell}$ represent the (unnormalized) logits and let us denote by $g^l(x) = h^l \circ \cdots \circ h^1(x) \in \mathbb{R}^{d_l}$ the $l$-th layer features. Traditional post-hoc OOD detection methods can be split into two main families:

**Logit-based.** The OOD score $s : \mathcal{X} \to \mathbb{R}$ is obtained by applying a scoring function $\sigma : \mathbb{R}^{d_\ell} \to \mathbb{R}$ on the logit space: $s = \sigma \circ f$.

**Feature-based.** The OOD score $s : \mathcal{X} \to \mathbb{R}$ is obtained by applying a scoring function $\sigma : \mathbb{R}^{d_l} \to \mathbb{R}$ on an intermediate feature space: $s = \sigma \circ g^l$ for some $1 \leq l < \ell$. The most common choice is to take the penultimate layer features, i.e. $l = \ell - 1$.

Moreover, in the OOD literature, some techniques have been applied to some of the previous OOD detectors to augment their performances:

**Activation shaping**. These methods can be seen as *add-ons* to logit-based scores: an additional (clipping, normalization,...) transformation $\tau : \mathbb{R}^{d_{\ell-1}} \to \mathbb{R}^{d_{\ell-1}}$ is applied to the penultimate layer $g^{\ell-1}$ before applying the last layer. The scoring function becomes $s = \sigma \circ h^{\ell} \circ \tau \circ g^{\ell-1}$.

**Layer-wise score aggregation.** This method can be seen as an *add-on* to feature-based scores. It aggregates the scores computed on different internal layers. For a given example $x$, a set (or a subset)

---

[1]OODEEL: `https://anonymous.4open.science/r/oodeel-C750`
[2]Visualization platform: `https://oodeel-submission.streamlit.app/`

of $\ell - 1$ different scores are computed from the different feature spaces $s_l(x) = \sigma_l \circ g^l(x)$, which are then aggregated together into a single score via an aggregation function $\mathcal{A} : \mathbb{R}^{\ell-1} \to \mathbb{R}$, i.e. the final score is $s(x) = \mathcal{A}\big(s_1(x), \ldots, s_{\ell-1}(x)\big)$.

Note that some of the OOD methods require *fitting* the functions $\sigma$, $\tau$, or $\mathcal{A}$ on the ID training data, while others do not. We give references to such techniques in Section 3.3 for conciseness.

## 2.2 MOTIVATIONS FOR (ANOTHER) POST-HOC OOD DETECTION LIBRARY

Post-hoc OOD is convenient because it theoretically applies to any pre-trained model and does not need any specific training procedure. Moreover, it has been shown by the OpenOOD benchmark that this class of OOD detectors is on par with other training-based methods Zhang et al. (2023). Hence, some libraries such as OpenOOD and Pytorch-OOD implement a large set of post-hoc OOD detectors. However, **(1)** these libraries implement OOD detectors separately, as in the original papers, whereas some ideas from one could be applied to others. For instance, activation shaping techniques can be applied to all logit-based methods, and layer-wise score aggregation to all feature-based methods. In OODEEL, we use abstractions that allow us to apply these components to any OOD detector, when appropriate. Moreover, **(2)** they restrict the application of these methods to predefined models, which are manually overloaded one by one to be equipped with OOD detection capabilities. We believe that the strength of post-hoc OOD detection lies in its broad applicability to any pre-trained model. Hence, in OODEEL, we developed a tool to build OOD detectors on top of any user-provided pre-trained model. Finally **(3)** these libraries are developed in PyTorch only, missing the TensorFlow community. As we shall see, OODEEL's engine is built to make OOD detectors available for both backends, without duplicating code.

## 3 A LIBRARY FOR UNIFIED POST-HOC OOD DETECTION

In this section, we give an overview of OODEEL's usage and features. We first emphasize the simplicity of its API and how to use it for any model in both TensorFlow and PyTorch. Then, we describe its core abstractions and how they enable a unified and broad application of Post-hoc OOD detection. We also provide a list of implemented OOD detectors. Finally, we describe some additional quality-of-life features that are bundled in OODEEL. All the described features are explained in detail, together with API documentation and Jupyter tutorials, in OODEEL's documentation.

## 3.1 USER API

The basic workflow of using OODEEL can be broken down into three main steps:

```python
from oodeel.datasets import load_data_handler

data_handler = load_data_handler("torch")

# 1a- Load in-distribution dataset: CIFAR-10
ds_fit = data_handler.load_dataset(
    "CIFAR10", load_kwargs={"root": data_path, "train": True, "download": True}
)
ds_in = data_handler.load_dataset(
    "CIFAR10", load_kwargs={"root": data_path, "train": False, "download": True}
)
# 1b- Load out-of-distribution dataset: SVHN
ds_out = data_handler.load_dataset(
    "SVHN", load_kwargs={"root": data_path, "split": "test", "download": True}
)

# 2- Prepare data (preprocess, shuffle, batch)

# define preprocessing function "preprocess_fn"
# define batch_size

ds_fit = data_handler.prepare(
    ds_fit, batch_size, preprocess_fn, shuffle=True, columns=["input", "label"]
)
ds_in = data_handler.prepare(
    ds_in, batch_size, preprocess_fn, columns=["input", "label"]
)
ds_out = data_handler.prepare(
    ds_out, batch_size, preprocess_fn, columns=["input", "label"]
)
```

```python
from oodeel.methods import Mahalanobis
from oodeel.aggregator import StdNormalizedAggregator

mahalanobis = Mahalanobis(
    eps=0.05,
    aggregator=StdNormalizedAggregator()  # alternatively,
)  # isherAggregator()
mahalanobis.fit(
    model,
    feature_layers_id=[
        "layer1.2.relu",
        "layer2.2.relu",
        "layer3.2.relu",
    ],
    fit_dataset=ds_fit,
)
scores_in, _ = mahalanobis.score(ds_in)
scores_out, _ = mahalanobis.score(ds_out)

# === metrics ===
# auroc / fpr95
metrics = bench_metrics(
    (scores_in, scores_out),
    metrics=["auroc", "fpr95tpr"],
)
```

Figure 1: User API for using Mahalanobis detector with CIFAR-10/SVHN as ID/OOD datasets. **(1) Data preparation:** Instantiate a `DataHandler` depending on the backend, load the ID dataset from the default hub. **(2) Detector instantiation and scoring:** Instantiate the detector, fit it to the user-provided pretrained classifier, and score the datasets. **(3) Compute OOD metrics**.

1 **Data preparation:** Load and prepare the dataset using the `DataHandler` object. A TensorFlow `tf.data.Dataset` or a PyTorch `torch.data.Dataset` is

loaded either from the pre-specified hub (`tensorflow-datasets` for TensorFlow and `torchvision` for PyTorch by default, but both can load data from Hugging-Face with the argument `hub=huggingface`), or from custom input data (`tuple` or `dict` of `np.ndarray`, or existing TensorFlow or PyTorch `Dataset`). Then, `data_loader.prepare()` prepares the Dataset for scoring, and returns a prepared `tf.data.Dataset` or a `torch.data.DataLoader`.

2 **OOD Detector instantiation and scoring:** Instantiate the detector with appropriate hyper-parameters depending on the chosen detector. Note that all the logit-based detectors can be used with implemented activation-shaping techniques, ReAct, ASH, and SCALE, with the arguments `use_react/ash/scale=True`. Then, fit the detector to the provided `keras` or `torch.nn` pretrained classifier. Feature-based detectors (like DKNN or Maha-lanobis) leverage internal representations of neural network classifiers. For these detectors, the user has to indicate what layer(s) the detector will use to compute the score through the argument `feature_layers_id`, which is a list of names of such layers. The names can be easily found using `.summary()` functions or convenient graph visualization tools such as Netron app. When several layers are given as input, the detector aggregates the scores for each layer using an `aggregator` object. Often, these feature-based detectors need a third ID fit dataset, which can be built out of the ID training dataset. In such cases, the user needs to specify it with the argument `fit_dataset`.

3 **Computing OOD metrics:** Compute the selected OOD metrics from the scores computed in the previous step. Note that it is possible to give any metrics from `sklearn.metrics` as an argument when relevant.

**The APIs are the same for TensorFlow and Pytorch**, from data loading to computing metrics. All these steps are illustrated for the Mahalanobis detector in Figure 1.

## 3.2 CORE STRUCTURE

OODEEL is built on four main core abstractions we outline in this Section. The interactions between these abstractions for a detection workflow are illustrated in Figure 2.

**DataHandler** From the user's point of view, this abstraction instantiates data from dataset hubs Torchvision, Tensorflow Datasets, or Hugging Face, or from user-provided datasets. It also pre-pares datasets for scoring (prepared datasets can also be used for training). It is instantiated using `load_data_handler()`, which either loads `TFDataHandler` or `TorchDataHandler` au-tomatically depending on the backend. The two child classes share the same API, which does not depend on the backend, so all this is transparent to the user.

**FeatureExtractor** This abstraction runs completely under the hood, and extracts internal features of any pretrained models in `keras` (with `KerasFeatureExtractor` child class) or `torch.nn` (with `TorchFeatureExtractor` child class) based on layer names or indices. It is fit on a model when using `OODBaseDetector.fit()` and can be used in place of the original model. Its `.predict()` method returns the model outputs with its internal features.

**Operator** This abstraction also runs under the hood and provides a single API for tensor opera-tions, regardless of the backend. When instanciating a detector, the backend is detected and either `TorchOperator` or `TFOperator` is loaded. All the detectors' calculations are implemented with the `Operator`'s API and hence do not need a separate implementation to work with PyTorch or Tensorflow. The spirit of this abstraction is the same as `keras.ops`[3] for Keras 3, or EagerPy Rauber et al. (2020).

**OODBaseDetector** It is the main engine of the library, and all detectors inherit from this class. It manages the use of activation-shaping and layer aggregation techniques, and loads appropriate `DataHandler`, `FeatureExtractor`, and `Operator` that will be used at runtime.

Some key practical advantages of the OODEEL implementation deserve to be highlighted.

**Compatibility with both PyTorch and Tensorflow:** The implementation of a common API for PyTorch and TensorFlow specific classes allows us to write code using this API, without worrying about which class, and therefore which backend will be used.

---

[3]`https://keras.io/api/ops/`

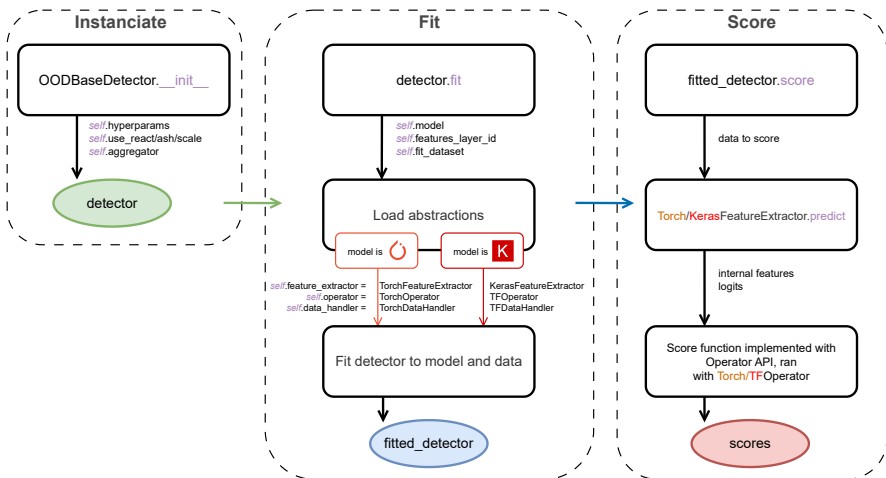

Figure 2: Details on interactions between `OODBaseDetector`, `DataHandler`, `FeatureExtractor` and `Operator` when using OODEEL. Note that for simplicity, we omit to represent `HFFeatureExtractor`

**Seamless data loading:** Using `DataHandler`, data can be loaded from main datasets hubs: Torchvision, Tensorflow Datasets, and HuggingFace datasets, or from custom datasets, with a backend-agnostic API.

**Compatibility with any pretrained Keras and torch.nn classifier models:** `FeatureExtractor` and its child classes allow the extraction of internal features from all pretrained models solely using the layer names. This is a key advantage compared to other OOD libraries that reimplement common models to make them compatible with their code.

**Unified implementation:** `OODBaseDetector` is implemented so that, when it is applicable, every detector's component that can be used with other detectors, even if it was not the case in these detectors' initial implementation. Hence, we can apply activation shaping techniques such as ReAct to every logit-based detector, and aggregate the scores of several layers for all feature-based detectors.

**Easy implementation of new custom detectors:** Using `Operator`, it takes only a few class overrides to implement a new detector that enjoys all the previous features. A tutorial in the documentation is provided to guide practitioners through such an implementation.

### 3.3 IMPLEMENTED OOD DETECTORS

OODEEL is not to be compared with benchmarking software like OpenOOD. It implements fewer OOD detection techniques since our focus is to unify their implementations and make them applicable to any model in both TensorFlow and PyTorch. Hence, we selected OOD detection techniques that are either the most cited or among the best of the OpenOOD benchmark. The included OOD detection techniques, sorted by category, are as follows. **Feature-based detectors**: Mahalanobis Lee et al. (2018), DKNN Sun et al. (2022), VIM Wang et al. (2022), RMDS Ren et al. (2021), SHE Zhang et al. (2022), Gram Sastry & Oore (2020). **Logit-based detectors**: MLS Vaze et al. (2022), MSP Hendrycks & Gimpel (2018), Energy Liu et al. (2021), Entropy Ren et al. (2019), GEN Liu et al. (2023), ODIN Liang et al. (2020). **Activation-shaping techniques**: ReAct Sun et al. (2021), ASH Djurisic et al. (2022), SCALE Xu et al. (2023). **Layer-wise aggregation techniques**: Fisher Dadalto et al. (2024); Haroush et al. (2022), Normalized Sastry & Oore (2020).

### 3.4 ADDITIONAL FEATURES

OODEEL comes with additional visualization and benchmarking features that we describe in the Appendix. We use the benchmarking feature in the following section.

## 4 A Unified Benchmark for Post-hoc OOD Detection

OpenOOD is the reference benchmark for OOD detection. However, the leaderboard they provide suffers from limitations. **(1)** No generic feature aggregation techniques are tested with feature-based detectors. **(2)** Activation shaping is only applied to energy, whereas it could be applied to any logit-based methods. These two limitations are design choices that are perfectly legitimate when the intention is to provide a benchmark faithful to the methods presented in the original papers. **(3)** More problematically, each experiment is run on very few different neural networks. As an example, OOD detectors are tested on CIFAR-10 and CIFAR-100 with only one model, and on ImageNet with three models, whereas, as we shall see, the OOD detector's performance rank can change a lot depending on the underlying model.

Motivated by these limitations, we leverage the holistic and interoperable implementation of OODEEL OOD detectors to comprehensively test the **impact of layer-wise score aggregation**, **the impact of activation shaping**, and **the effect of the model** (including on activation shaping and layer-wise aggregation). We also leverage our experiments to provide additional insights into the link between ID accuracy and AUROC, disparate performances on small-scale and large-scale datasets, and correlation between near- and far-OOD.

### 4.1 Setting

We test all the implemented OOD detectors following OpenOOD benchmark settings, with CIFAR-10, CIFAR-100, and ImageNet and Imagenet full-spectrum as ID datasets and corresponding near-OOD and far-OOD sets of OOD datasets. For CIFAR-10, CIFAR-100, we test the detectors on 11 different models, and for ImageNet, on 7 different models. We select the best hyperparameters for each OOD detector using a simple grid search. Logit-based detectors are all assessed with and without activation-shaping, and feature-based detectors with three levels of score aggregations: no aggregation, partial aggregation, where we aggregate the last layers, and full aggregation, where we aggregate all the layers. We use the Fisher aggregation technique. As a result, a total of 42 different detectors are tested on 29 different models. All the following figures are extracted from our visualization platform https://oodeel-submission.streamlit.app/. The full results of our experiments can be found in the Appendix and in the platform. Note that Imagenet and Imagenet full-spectrum results convey the same conclusions, so we leave Imagenet full-spectrum in the platform only. In addition, we only report AUROC in the present manuscript, but other metrics such as FPR@95TPR and AC can be browsed in the website - they also convey the same insights. Below we give a summary of the main results and takeaways by displaying only a subset of the results, for the sake of readability.

### 4.2 Impact of the model

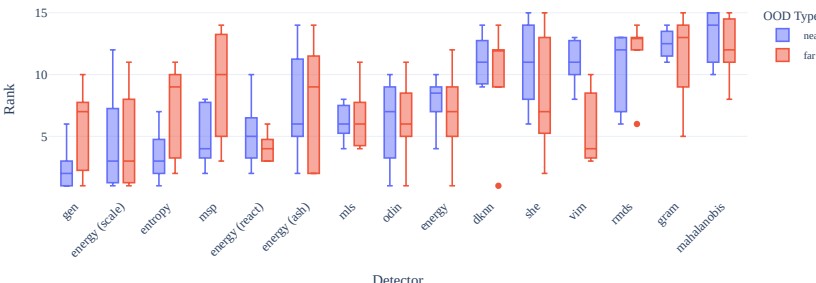

Figure 3: Box plots of the rank for each detector on ImageNet. Methods are sorted according to their mean rank on near-OOD.

For each model, we rank the different OOD detectors according to their AUROC. We then produce a box plot of these ranks with respect to the models for each OOD detector. The results of this experiment for the ImageNet benchmark and the most common OOD detectors are reported in Fig. 3. As we can see, the rank varies greatly with the model, and there is no clear winner: for the near-OOD experiments, GEN is the best detector on average, but DKNN is the best detector for one particular model while it ranks 10th on average. We also provide rank correlation heatmaps between OOD detectors for each models in Appendix C to corroborate this finding.

**Takeaway 1**   Results on OOD detectors reported on only one or few models should be interpreted with care, and might not be representative of the real behavior on a wider range of models.

## 4.3   IMPACT OF LAYER-WISE SCORE AGGREGATION

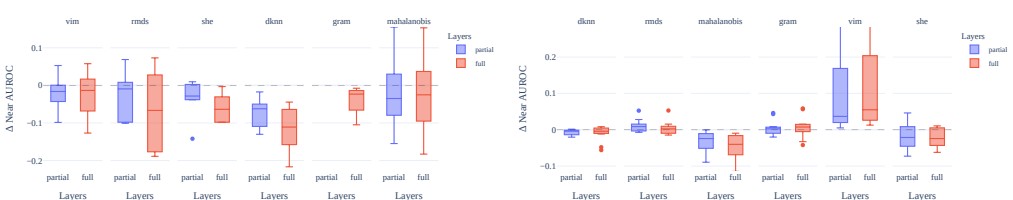

Figure 4: Box plot of the $\Delta$ in near-OOD AUROC between feature-based detectors applied on the penultimate layer and either a subset of the internal layers (partial) or all internal layers (full). **Left:** ImageNet, **Right:** CIFAR-10.

Several recent papers suggest building OOD detectors by aggregating feature-based OOD scores computed layer-wise. We illustrate the impact of layer-wise score aggregation with the following experiment. For each model and feature-based OOD detectors, we compute the gain/loss in AUROC from using layer-wise aggregation with respect to the vanilla version (i.e. computed from the penultimate layer only). For each OOD detector, we produce the box plot of these gains/losses for different levels of aggregation (partial and full) with respect to the models. The results on ImageNet and CIFAR-10 near-OOD are reported in Fig. 4. For ImageNet, score aggregation can improve the AUROC but hurts more often than it helps. For CIFAR-10, the results are more positive: it helps more often than it hurts, and greatly boosts VIM's detection capabilities.

**Takeaway 2**   Score aggregation can have a positive impact, but this positive impact is not systematic. For small-scale tasks, it helps most of the time, but for large-scale tasks, it tends to deteriorate the performance for most models.

## 4.4   IMPACT OF ACTIVATION SHAPING

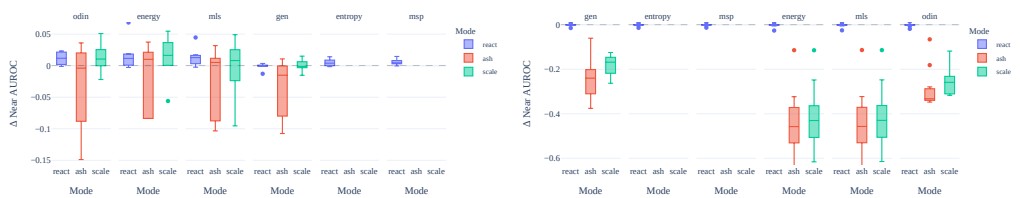

Figure 5: Box plot of the $\Delta$ in near-OOD AUROC between logit-based detectors and their augmentation with activation-shaping techniques (ReAct, SCALE, and ASH). **Left:** ImageNet, **Right:** CIFAR-10.

Activation shaping techniques transform the penultimate layer features before applying the last linear layer and computing logit-based scores. In the original papers of the ReAct, ASH and SCALE methods, results are reported by combining these activation shaping techniques with the Energy logit-based score alone. In order to evaluate the impact of activation shaping on other logit-based scores, we perform the following experiment. For each model and feature-based OOD detector, we compute the gain/loss in AUROC from using activation shaping with respect to the vanilla version, i.e. without activation shaping. For each OOD detector, we produce the box plot of these gains/losses for React, ASH and SCALE with respect to the models. The results on ImageNet and CIFAR-10 near-OOD are reported in Fig. 5. As we can see, activation shaping is mostly beneficial on ImageNet: it is systematically so for the ReAct method, while for ASH and SCALE greater care should be taken, as there are models for which they have negative effects. These results are in great contrast to those obtained for CIFAR-10: ASH and SCALE have a nefarious effect on most logit-based scores and models, losing up to 0.6 in AUROC, while the effect of using ReAct is negligible.

**Takeaway 3**   Using activation shaping helps on large-scale tasks (for ReAct, consistently, and for ASH and SCALE, more randomly). For small-scale tasks, it has little effect in most cases, while SCALE and ASH have the potential to greatly damage the original detectors.

## 4.5   ADDITIONAL INSIGHTS

### 4.5.1   LINK BETWEEN ID ACCURACY AND OOD DETECTION PERFORMANCES

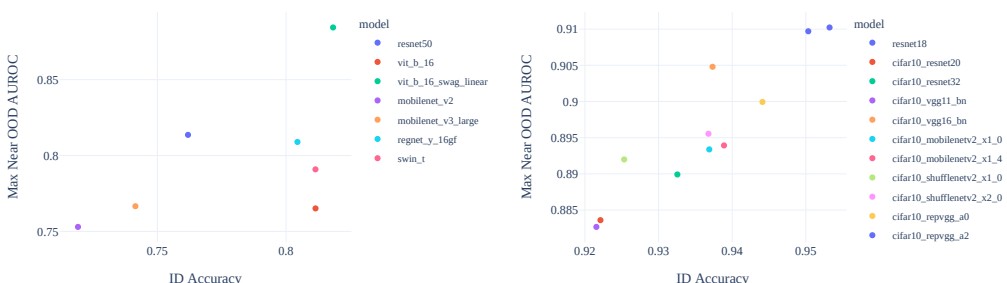

Figure 6: Scatter plot with in-distribution accuracy on x-axis and near-OOD AUROC on y-axis. Each point is a model, whose assigned AUROC is the best among all OOD detectors applied to this model. **Left:** ImageNet, **Right:** CIFAR-10.

OODEEL allows for a thorough evaluation of ID accuracy versus OOD detection performance across different neural network architectures. In Fig. 6, we report the results of testing multiple different model architectures on the ImageNet and CIFAR-10 near-OOD benchmarks. For each model architecture, the AUROC of the best OOD detector (for that model and setting) is reported. As we can see, ID accuracy and OOD detection are linearly correlated in the case of CIFAR-10, while there is no clear pattern emerging from the ImageNet experiment: the four best models achieve similar ID accuracy while having very different OOD detection performances. This conclusion contrasts with Vaze et al. (2022), which observed that improving ID accuracy was an efficient way of enhancing OOD detection performances.

**Takeaway 4**   Correlation between ID accuracy and OOD detection seems to be near linear for small-scale tasks and absent for large-scale tasks.

### 4.5.2   DISPARATE PERFORMANCES FOR DETECTORS ON SMALL-SCALE AND LARGE-SCALE TASKS

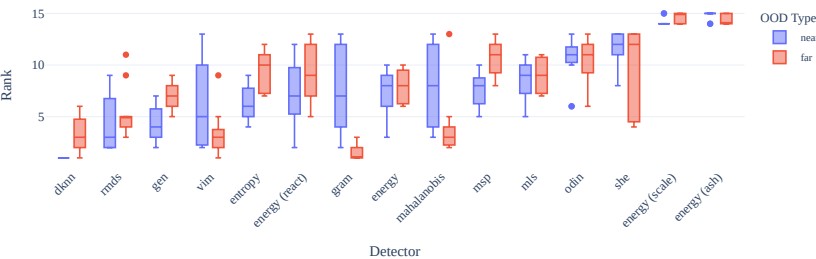

Figure 7: Box plots of the rank with standard deviation for each method (OOD detector) on CIFAR-10. Methods are sorted according to their mean rank on near-OOD.

By comparing the ranks of the different detectors on ImageNet as displayed in Fig. 3 with those obtained by performing the equivalent experiment on CIFAR-10 (Fig. 7), we can conclude that logit-based detectors perform better for large-scale models while feature-based detectors perform better for small-scale tasks. This might explain why activation shaping dedicated to logit-based helps for ImageNet but not for CIFAR-10, and conversely for feature aggregation.

**Takeaway 5**  Logit-based detectors perform better on large-scale tasks, and feature-based detectors perform better on small-scale tasks.

### 4.5.3  TRADE-OFF BETWEEN NEAR-OOD AND FAR-OOD

We can also observe from Fig. 8 that for some ID datasets, there is a tradeoff between performing well on the near-OOD datasets and performing well on the far-OOD datasets. This is particularly clear in the case of the CIFAR-100 dataset, and even if less so, it remains true for ImageNet and CIFAR-10.

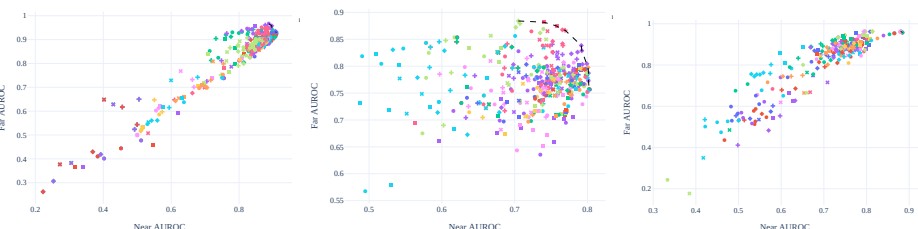

Figure 8: Scatter plot with near-OOD AUROC on x-axis and far-OOD AUROC on y-axis. EAch point is a pair model / OOD detector. **Left:** CIFAR-10, **Middle:** CIFAR-100, **Right:** ImageNet.

**Takeaway 6**  For some tasks, near-OOD and far-OOD might not be correlated.

> **Overall Takeaway**
>
> Perhaps the most important takeaway of this work is that in Post-Hoc OOD detection, detector performances greatly depend on the task (dataset and model). Benchmarking their performance appropriately requires testing it on at least several models, which is not classically done in post-hoc OOD detection research. But it can at most mitigate the issue. For post-hoc OOD detection on real-world applications, i.e., with user-specific datasets and models, it is crucial to be able to test detectors on the specific task at hand, hence the value of OODEEL, which can be applied to any model and dataset.

## 5  CONCLUSION AND FUTURE WORKS

We introduced OODEEL, a holistic and extensible library for post-hoc OOD detection, designed to work seamlessly across PyTorch and TensorFlow models. By unifying key components—such as activation shaping and layer-wise score aggregation—OODEEL provides a powerful platform for both practical application and rigorous experimentation. Our large-scale benchmarking reveals important nuances in post-hoc OOD detection performance, including the impact of model choice, the varying utility of aggregation and activation shaping techniques, the context-dependent relationship between ID accuracy and OOD detection efficacy, the disparate performances of OOD detectors on different tasks and on near-OOD and far-OOD detection. These findings emphasize the need for careful, context-aware evaluation of OOD detectors. With its open-source release and accompanying benchmarking interface, OODEEL aims to support reproducible, extensible, and insightful OOD research for the broader machine learning community.

While OODEEL offers a comprehensive and modular framework for post-hoc OOD detection, some areas of improvement remain. First, the library currently implements a subset of the most cited and best-performing OOD detectors from OpenOOD, prioritizing compatibility and unification over exhaustiveness. Consequently, newer or more experimental detectors may not be readily available. Second, although our benchmark expands beyond existing evaluations by testing across many models and configurations, it remains limited to image classification tasks. Since OODEEL's `FeatureExtractor` class can be applied to any model, tackling more complex tasks such as object detection or semantic segmentation is the most direct follow-up of our work.

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

## A   APPENDIX: OODEEL'S VISUALIZATION AND BENCHMARKING FEATURES

**Visualization**   OODEEL also comes with quality-of-life features. The first feature includes three useful visualizations that are often used in OOD detection research: Histogram plots for ID and OOD scores, ROC curve, and t-SNE plot of the model's penultimate layer. The first two are used to assess the performance of a detector, and the last is often used to qualitatively evaluate the discriminating capabilities of a given neural network, which often prefigures its Post-hoc OOD detection performances. Examples of these visualizations are found in Figure 9

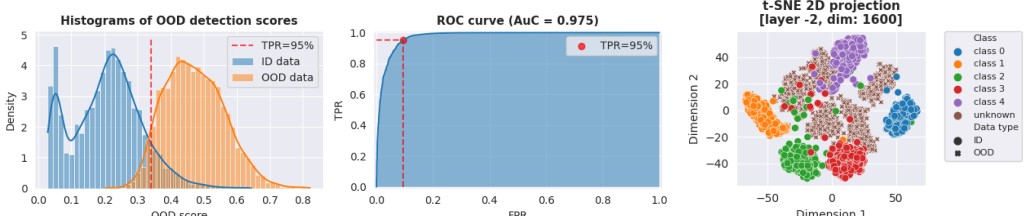

Figure 9: Visualization tools provided by Oodeel. **From left to right:** Histograms and KDE plots for ID vs OOD scores, ROC curve, and t-SNE visualization of a model's feature space.

**Benchmarking**   The last feature is a benchmarking utility that allows for easy testing of the performance of OODEEL's detectors on OpenOOD benchmark. This feature comes with a clean configuration logic, checkpointing, and sharding capabilities, reducing the benchmarking hassle. It also records computational performance metrics, such as VRAM utilisation and execution time. The experiments can be tracked from the terminal through a `rich` interface, and logged to WeightsAndBiases (illustration of the interface in Figure 10). The benchmarking utility is part of another repository called oodeel-banchmark[4] hosted on GitHub.

```
[11:40:44] Running [uid= df16d2cc] → cifar10 / cifar10_resnet20 / odin(react):none
[11:40:56] [mem] Fit              Δ=    0.0 MiB  peak=   40.4 MiB  time= 11.4s
[11:40:59] [mem] ID score         Δ=    0.0 MiB  peak=  213.4 MiB  time=  3.2s
[11:41:03] [mem] OOD cifar100      Δ=    0.0 MiB  peak=  213.4 MiB  time=  3.5s
[11:41:07] [mem] OOD tin           Δ=    0.0 MiB  peak=  213.4 MiB  time=  3.0s
[11:41:22] [mem] OOD mnist         Δ=    0.0 MiB  peak=  213.4 MiB  time= 14.9s
[11:41:36] [mem] OOD svhn          Δ=    0.0 MiB  peak=  213.4 MiB  time=  7.0s
[11:41:42] [mem] OOD texture       Δ=   -0.0 MiB  peak=  213.4 MiB  time=  4.6s
[11:41:52] [mem] OOD places365     Δ=    0.0 MiB  peak=  213.4 MiB  time=  9.4s
[11:41:54] ✓ cifar10 | cifar10_resnet20 | odin:none → AUROC-near=0.820  AUROC-far=0.823  HM=0.821
```

Figure 10: Caption

## B   APPENDIX: DETAILS ON THE BENCHMARK SETTING

In this appendix, we provide a complete description of the hardware, software, datasets, models, and orchestration logic used in our out-of-distribution (OOD) detection benchmark. Our goal is to enable exact reproduction of all experiments.

### B.1   COMPUTE ENVIRONMENT

We ran all experiments on a single machine equipped with:

**Hardware**

- 2 × NVIDIA RTX 4090 GPUs (24 GB VRAM each)
- CPU RAM: 128 GB

**Software**

---

[4] https://github.com/deel-ai/oodeel-benchmark

- Python 3.10, PyTorch 2.7.0, torchvision 0.22.0
- OODeel library (dev branch, commit `ff4fa17`, will soon be merged)

**Runtime & Logging**

- Wall-clock time: $\approx 1$ week on $2 \times$ RTX 4090
- Batch size: 64 per GPU; peak VRAM $\leq 20$ GB
- Metrics streamed to Weights & Biases (and available in "All runs" leaderboard from `https://oodeel-benchmark.streamlit.app/`)

## B.2 REPOSITORY & BENCHMARK ORCHESTRATION

All code is available at `https://github.com/deel-ai/oodeel-benchmark`. We drive the full sweep with:

```
python -m src.run --num-shards 2 --shard-index 0
```

This script:

1. Loads every YAML in `configs/datasets/`, `configs/models/`, `configs/methods/`.
2. Forms the Cartesian product of (dataset, model, method, hyperparameters) configurations.
3. Shards the list via `-num-shards`/`-shard-index`.
4. Skips runs with existing `.parquet` outputs in `results/`.
5. Streams raw scores, AUROC, and TPR@5% FPR to W&B.

## B.3 SWEEP CONSTRUCTION & CONFIG LOGIC

Each YAML file defines a flat key–value mapping:

- `configs/datasets/*.yaml`: list of near- and far-ood datasets for a given id dataset, model names.
- `configs/models/*.yaml`: architecture name, checkpoint source, feature-extraction layers.
- `configs/methods/*.yaml`: detector type and hyperparameter grid.

The driver merges one file from each folder into a single run configuration. Sharding evenly slices this list across processes, and checkpointing ensures idempotent restarts.

## B.4 DATASETS & MODELS TESTED

We follow the OpenOOD benchmark in pairing each in-distribution (ID) dataset with semantically *near* and *far* OOD test sets.

**CIFAR–10:**
- **Near-OOD:** CIFAR–100, Tiny ImageNet
- **Far-OOD:** MNIST, SVHN, Textures, Places365

**CIFAR–100:**
- **Near-OOD:** CIFAR–10, Tiny ImageNet
- **Far-OOD:** MNIST, SVHN, Textures, Places365

**ImageNet–1k:**
- **Near-OOD:** SSB-hard, NINCO
- **Far-OOD:** iNaturalist, Textures, OpenImage-O

**Model Architectures**   We evaluate 11 models on CIFAR and 7 on ImageNet, including:

- **ResNets:** ResNet-18, ResNet-32, ResNet-50
- **VGG:** VGG11, VGG16, RepVGG-A0, RepVGG-A2

- **MobileNets:** MobileNet v2, MobileNet v3-Large
- **ShuffleNets:** ShuffleNet v2
- **Transformers:** ViT-B/16, Swin-Tiny
- **RegNet:** RegNetY-16GF

## B.5   OOD Detectors & Hyperparameter Sweeps

We evaluate 12 base OOD detection methods (e.g. MSP, Energy, Mahalanobis, ODIN, DKNN, SHE, VIM, . . . ) and for each base method construct variants as follows:

- **Logits-based variants:** with and without ReAct clipping, ASH, and SCALE transformations.
- **Feature-based variants:** with and without score aggregation over different feature layers.

This yields a total of 42 distinct method configurations. All hyperparameter search grids are specified in the `configs/methods/` folder of the oodeel-benchmark repository.

## B.6   Logging, Results & Reproducibility

Each run writes a Parquet file in `results/` containing raw scores, AUROC, and TPR@5% FPR. Visualizations are provided via console bars (Rich) and W&B dashboards. All code, exact commit hashes, and example launch commands (including seed settings in `src/utils.py`) are publicly available, ensuring full reproducibility.

# C APPENDIX: COMPLEMENTARY BENCHMARK RESULTS

In this section, we include complementary plots to the main document for CIFAR-100 and Far-OOD experiments. This section is structured as the experiments section.

## C.1 IMPACT OF THE MODEL

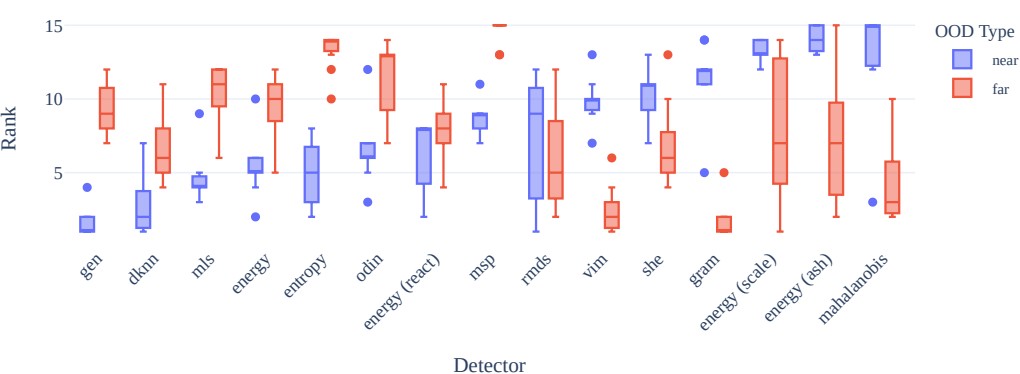

Figure 11: Box plots of the rank for each detector on CIFAR-100. Methods are sorted according to their mean rank on near-OOD. For CIFAR-100, the rank is very different for Near-OOD and Far-OOD, which corroborates the observation of the main paper that performances on Near and Far OOD do not always correlate.

We also report the Spearman rank correlation heatmap between the detectors' rank per model in Figure 12 for CIFAR-10, CIFAR-100, and ImageNet. We can see that there is some correlation in the rank of the detectors for CIFAR-10 and CIFAR-100, even if it is not high enough to alleviate the need to test every detector on various models. For ImageNet, the correlation is significantly lower, emphasizing this need even strongly.

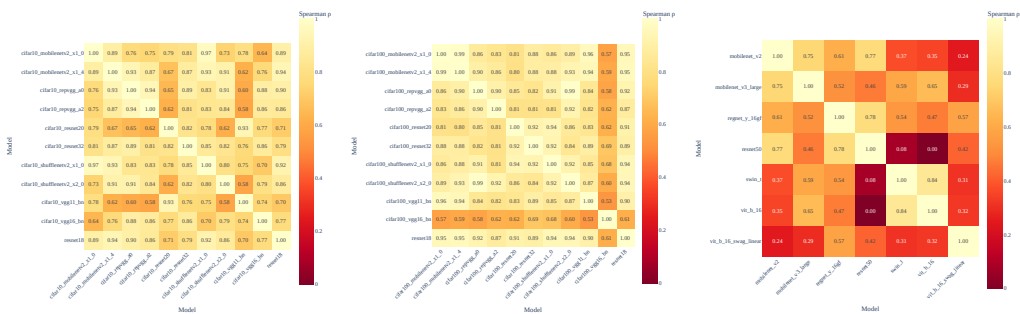

Figure 12: Spearman rank correlation between the AUROC of the detectors for each model. **From left to right:** CIFAR-10, CIFAR-100, ImageNet.

## C.2 IMPACT OF LAYER-WISE SCORE AGGREGATION

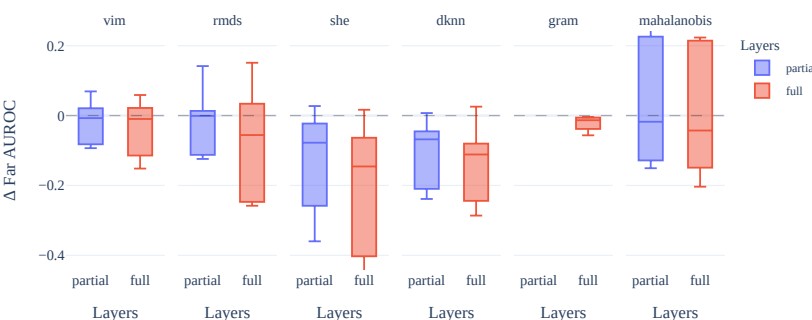

Figure 13: Box plot of the $\Delta$ in AUROC between feature-based detector applied on the penultimate layer and either a subset of the internal layers (partial) or all internal layers (full) for **ImageNet Far-OOD**.

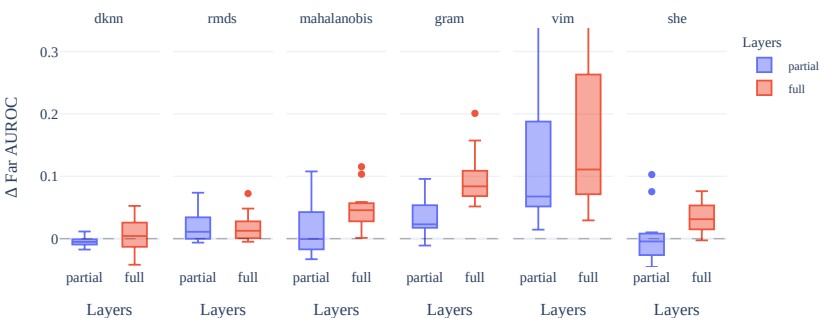

Figure 14: Box plot of the $\Delta$ in AUROC between feature-based detector applied on the penultimate layer and either a subset of the internal layers (partial) or all internal layers (full) for **CIFAR-10 Far-OOD**.

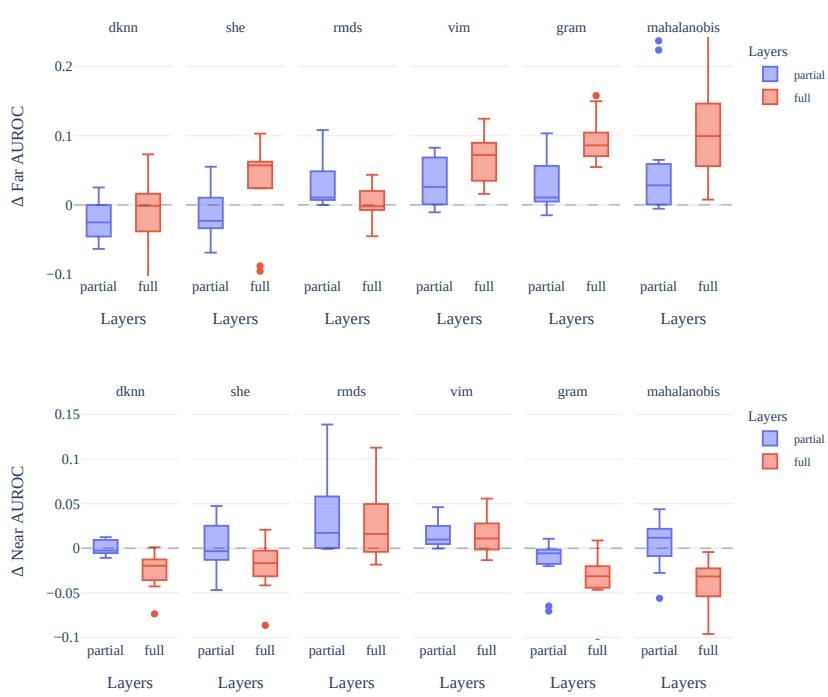

Figure 15: Box plot of the Δ in AUROC between feature-based detector applied on the penultimate layer and either a subset of the internal layers (partial) or all internal layers (full) for **top**: CIFAR-100 Near-OOD, **bottom**: CIFAR-100 Far-OOD

## C.3  IMPACT OF ACTIVATION SHAPING

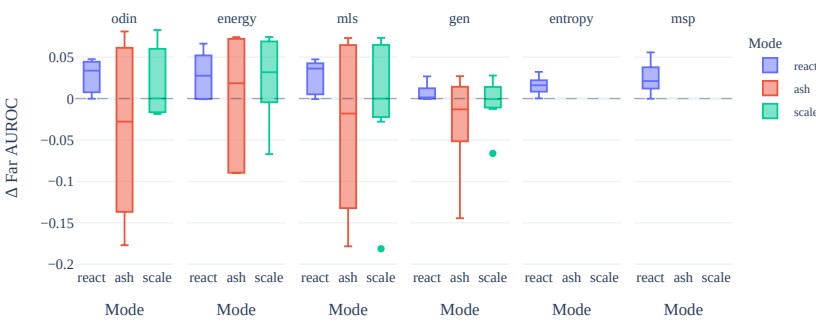

Figure 16: Box plot of the Δ in AUROC between logit-based detectors and their augmentation with activation-shaping techniques (ReAct, SCALE, and ASH) for **ImageNet Far-OOD**.

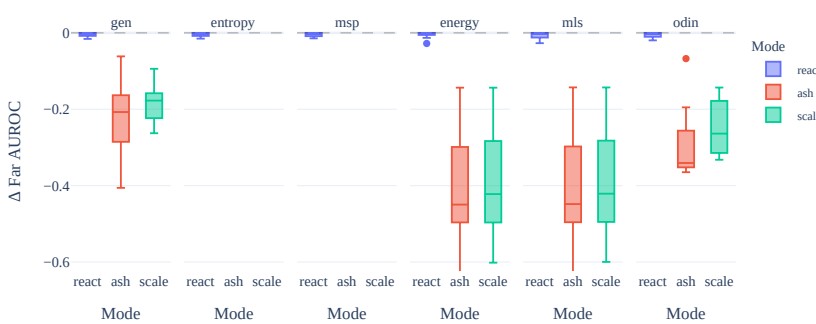

Figure 17: Box plot of the $\Delta$ in AUROC between logit-based detectors and their augmentation with activation-shaping techniques (ReAct, SCALE, and ASH) for **CIFAR-10 Far-OOD**.

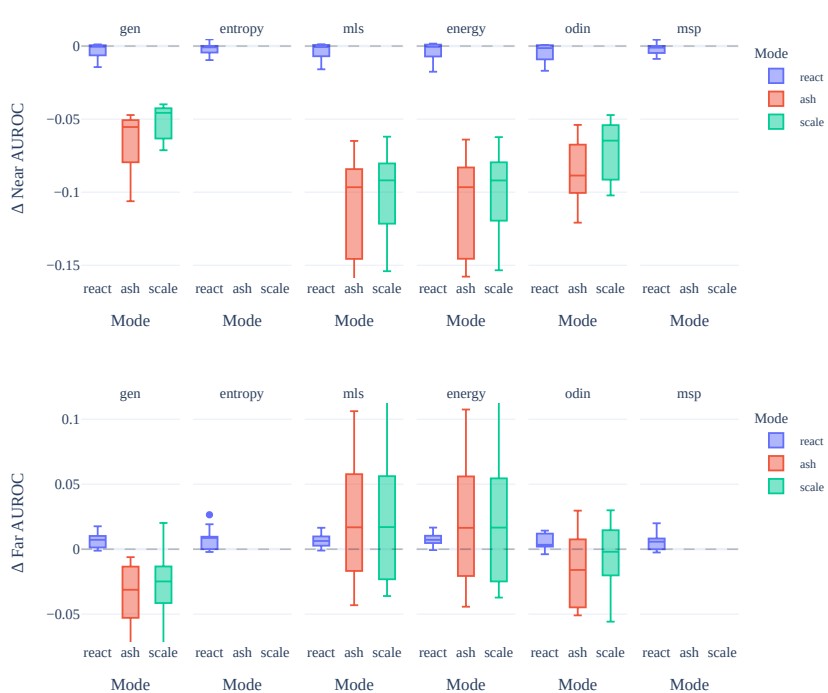

Figure 18: Box plot of the $\Delta$ in AUROC between logit-based detectors and their augmentation with activation-shaping techniques (ReAct, SCALE, and ASH) for **top**: CIFAR-100 Near-OOD, **bottom**: CIFAR-100 Far-OOD. Interestingly, the effect of activation shaping on Far-OOD is quite beneficial for CIFAR-100 Far-OOD, whereas it is not for CIFAR-100 Near-OOD or CIFAR-10 Far-OOD.

## D    APPENDIX: SANITY CHECK OF OODEEL DETECTORS IMPLEMENTATION BY PERFORMANCE COMPARISON WITH OPENOOD DETECTORS

To ensure the reliability of our OOD detection results using OODEEL, we conducted a sanity check by comparing our outcomes with those reported on the OpenOOD leaderboard. This validation step focused on both near-OOD detection tasks presented in Table 1.

Note that we obtain a significant improvement for the Gram method from OODEEL with respect to OpenOOD. It stems from a key implementation detail: in OODEEL, the Gram score relies on computing deviations based on quantiles (e.g., the 1st and 99th percentiles) of the features, whereas

OpenOOD uses the absolute minimum and maximum feature values. The use of quantiles in OODEEL leads to a more robust formulation for outliers.

Globally, the obtained AUROC aligns with OpenOOD. The differences are mainly slight, with OODEEL doing better than the OpenOOD implementation for 22 cases, equal for 4, and worse for 16 baselines. For a few exceptions, the difference is significant:

- OODEEL performs significantly better for Gram (three datasets), ODIN (CIFAR-10), and SHE (CIFAR-10)
- OpenOOD performs significantly better for RMDS (CIFAR-10)

Except for Gram, when there is such a large discrepancy, it is only for one dataset. It provides confidence in the correctness and robustness of our implementation.

Table 1: Comparison of **near-OOD** detection AUROC between OpenOOD and OODEEL. We used pretrained ResNet-18 for CIFAR-10 and CIFAR-100, and ResNet-50 for ImageNet.

| Method | CIFAR-10 | | CIFAR-100 | | ImageNet | |
|---|---|---|---|---|---|---|
| | OpenOOD | OODEEL | OpenOOD | OODEEL | OpenOOD | OODEEL |
| MLS | 87.52 | 87.89 | 81.05 | 79.89 | 76.46 | 76.46 |
| MSP | 88.03 | 88.30 | 80.27 | 79.05 | 76.02 | 76.02 |
| ODIN | 82.87 | 87.72 | 79.90 | 79.72 | 74.75 | 75.73 |
| GEN | 88.20 | 89.15 | 81.31 | 80.13 | 76.85 | 77.71 |
| SHE | 81.54 | 83.16 | 78.95 | 77.86 | 73.78 | 72.61 |
| Energy | 87.58 | 87.98 | 80.91 | 79.80 | 75.89 | 75.89 |
| ReAct | 87.11 | 87.93 | 80.77 | 79.75 | 77.38 | 77.39 |
| SCALE | 82.55 | 82.48 | 80.99 | 79.78 | 81.36 | 81.36 |
| ASH | 75.27 | 79.99 | 78.20 | 79.61 | 78.17 | 79.63 |
| DKNN | 90.64 | 91.02 | 80.18 | 79.92 | 71.10 | 72.24 |
| Mahalanobis | 84.20 | 84.91 | 58.69 | 59.43 | 55.44 | 55.83 |
| RMDS | 89.80 | 86.82 | 80.15 | 79.22 | 76.99 | 76.32 |
| VIM | 88.68 | 87.34 | 74.98 | 74.30 | 72.08 | 72.64 |
| Gram | 58.66 | 88.73 | 51.66 | 70.50 | 61.70 | 70.72 |

