# OpenReview forum: "OODEEL: A Holistic Library for Unified Post-Hoc OOD Detection Research And Application"
_ICLR.cc/2026/Conference — ICLR 2026 Conference Withdrawn Submission_

### Official Review · Reviewer_78G9 · 2025-10-22

**Soundness:** 3
**Presentation:** 2
**Contribution:** 3
**Rating:** 4
**Confidence:** 3

**Summary:**

> 1. The paper addresses three issues in existing post-hoc Out-of-Distribution (OOD) detection libraries: poor component interoperability (e.g., activation shaping and layer-wise aggregation techniques cannot be reused across detectors), limited model adaptability (only supporting predefined models), and single framework (mostly based on PyTorch).
> 2. To solve these issues, the authors propose OODEEL, an open-source library with a unified architecture built on four abstractions: DataHandler, FeatureExtractor, Operator, and OODBaseDetector. This architecture enables seamless compatibility with both PyTorch and TensorFlow frameworks, supports adaptation to any pre-trained model, and integrates key components such as activation shaping and layer-wise aggregation.
> 3. For validation, the authors conduct large-scale experiments. These findings confirm the value of OODEEL in unification, flexibility, and experimental support, providing a reproducible tool and benchmark for OOD detection research.

**Strengths:**

> 1. The design solves core issues, realizing cross-framework and cross-model compatibility through unified abstractions, enabling adaptation to both frameworks without additional coding.
> 2. Experimental design covers under-explored dimensions in the field: The paper addresses the long-neglected "impact of model variability" in OOD detection. It also verifies the effectiveness of layer-wise aggregation (partial/full aggregation) and activation shaping (ReAct/ASH/SCALE) across tasks of different scales. These experimental conclusions provide specific technical selection criteria for the field.
> 3. High practicality with comprehensive supporting resources: OODEEL provides a concise API (data preparation, detector instantiation, metric calculation) with consistent logic for both frameworks. visualization platform is publicly available, supporting real-time viewing of experimental results.

**Weaknesses:**

>  1. OODEEL only implements 12 basic detectors that are "highly cited or high-performing" from OpenOOD (e.g., Mahalanobis, MSP, Section 3.3). Additionally, core components such as activation shaping and layer-wise aggregation are integrations of existing technologies, with no original component design. So, it is recommended that future versions supplement new detectors to enhance the library's comprehensiveness.
>  2. It is impossible to prove the tool's effectiveness in non-classification tasks (Experimental scenarios are limited to image classification). It is suggested to supplement experimental cases of complex visual tasks to verify the library's generalizability in the future.
>  3. "partial aggregation" does not specify the specific number of "last layers". It is recommended to define the specific scope of aggregated layers to improve the rigor of expression.

**Questions:**

>-  1. Regarding the "Fisher technique" for layer-wise aggregation, the paper does not explain the logic of its parameter selection: Section 4.1 mentions that the Fisher technique is used for aggregation in all feature-based detectors, but it does not clarify why this technique was chosen over other aggregation methods. Could the authors supplement the selection "reasons" of this technique and details of hyperparameter settings?
>-  2. The negative effects of activation shaping technology on small-scale tasks lack in-depth analysis. The paper only describes the phenomenon but does not explain the mechanism (e.g., whether activation transformation destroys effective feature discriminability due to the concentrated feature distribution of small-scale datasets). Could the authors supplement the analysis?
>-  3. In practical applications, users often face private datasets. Could the authors explain the adaptation process of OODEEL on non-standard ID/OOD datasets and verify the tool's practical applicability?

---

### Official Review · Reviewer_CK9B · 2025-10-29

**Soundness:** 3
**Presentation:** 3
**Contribution:** 3
**Rating:** 2
**Confidence:** 3

**Summary:**

This paper introduces OODEEL, an open source, post hoc, model agnostic OOD detection library. It provides a unified framework that includes activation shaping and layer wise score aggregation, and it can be composed with many detectors through a simple and flexible API. The work also presents a comprehensive benchmark across multiple ID and OOD datasets, covering both near OOD and far OOD, to analyze robustness to model variability, the effects of layer wise aggregation and activation shaping, and the relationship between in distribution accuracy and OOD performance.

**Strengths:**

1. Proposes a unified, model-agnostic post-hoc OOD detection framework that is simple and flexible, supports many detectors, and facilitates practical use.

2. Covers multiple ID/OOD datasets, includes both near- and far-OOD settings, and evaluates across diverse architectures.

3. Provides detailed implementation descriptions, well-organized structure, and clear figures/tables with coherent analysis, making the work easy to understand and follow.

**Weaknesses:**

1. The work focuses solely on post-hoc OOD detection and evaluates only image classification. Consider adding at least one non-classification setting (e.g., detection/segmentation or text/audio classifiers) to support generality.

2. The set of layer-wise aggregation and activation-shaping techniques is limited. Compare broader alternatives (e.g., normalized or rank-based fusions, learned/convex blending, attention-weighted pooling; additional shaping/clamping schemes and percentile rules) and include sensitivity analyses over layer selections and hyperparameters.

3. Report computational cost (runtime, memory, throughput) for key configurations, with and without aggregation/shaping.

4. Expand the shift coverage to include additional distribution shifts (e.g., WILDS and other real-world shift benchmarks) to stress-test robustness across varied shift types.

**Questions:**

1. Provide a comprehensive table reporting OOD performance (e.g., AUROC, FPR95) and computational cost for both PyTorch and TensorFlow implementations, ideally with confidence intervals.

2. Discuss extensibility to detection/segmentation: clarify whether baseline post-hoc OOD techniques naturally extend to these tasks; include additional experiments or analyses to substantiate feasibility and limitations.

3. Detail pretrained data used for each model and analyze potential overlaps with OOD datasets; discuss how any overlap might bias results and include controls or alternative OOD sets where possible.

---

### Official Review · Reviewer_Qms2 · 2025-10-29

**Soundness:** 3
**Presentation:** 3
**Contribution:** 2
**Rating:** 4
**Confidence:** 3

**Summary:**

This paper presents OODEEL, an open-source library designed for unified, post-hoc OOD detection.
The library supports both PyTorch and TensorFlow backends and aims to provide a flexible and modular framework。
OODEEL abstracts common components such as data handling, internal feature extraction, activation shaping, and layer-wise score aggregation, enabling users to build, combine, and evaluate OOD detectors more efficiently.

**Strengths:**

- facilitates combining techniques like activation shaping (e.g., ReAct, ASH) and layer-wise feature aggregation across different OOD detectors, which are often siloed in existing implementations.

- The library is thoughtfully structured with clear abstractions (e.g., DataHandler, FeatureExtractor, Operator, OODBaseDetector), which promote reuse, modularity, and extensibility.

- Unlike existing libraries that are typically restricted to PyTorch, OODEEL supports both PyTorch and TensorFlow models through a unified API, reducing implementation burden for diverse user groups.

**Weaknesses:**

- The work primarily focuses on software engineering and system design rather than proposing new algorithms or theoretical insights for OOD detection.

- While the library enables studying several underexplored aspects of OOD detection (e.g., layer aggregation, activation shaping), the presented experiments mainly demonstrate usage rather than yielding new empirical or theoretical conclusions about OOD behavior.

- While OODEEL offers a solid and extensible platform for post-hoc OOD detection on image classifiers, The OOD detection community is increasingly shifting its attention toward multimodal settings (e.g., vision-language models like CLIP) and large language models (LLMs), where the definition and manifestation of distributional shifts can differ significantly from traditional image classification tasks. Extending OODEEL to support such architectures—or at least outlining a roadmap in that direction—would considerably strengthen its relevance.

**Questions:**

see Weakness

---

### Note · Authors · 2025-11-18

**Comment:**

We thank the reviewers for their time and constructive feedback. Their comments highlighted several important points for improving OODEEL, including expanding beyond image classification, broadening aggregation and activation-shaping methods, providing computational cost analyses, and outlining extensions toward multimodal and large-scale models.

We fully agree that these aspects require substantial revision. However, we believe that addressing them thoroughly and appropriately is not feasible within the rebuttal period. Therefore, we have decided to withdraw the submission.

We appreciate the reviewers’ helpful insights and will carefully incorporate their suggestions into a future, more complete version of the work.

**Withdrawal Confirmation:**

I have read and agree with the venue's withdrawal policy on behalf of myself and my co-authors.